# Diagnosis and Surgical Management of Insulinomas—A 23-Year Single-Center Experience

**DOI:** 10.3390/medicina59081423

**Published:** 2023-08-04

**Authors:** David Hoskovec, Zdeněk Krška, Jan Škrha, Pavol Klobušický, Petr Dytrych

**Affiliations:** 11st Department of Surgery, General University Hospital, U nemocnice 2, 12000 Prague, Czech Republicpetr.dytrych@vfn.cz (P.D.); 21st Medical Faculty, Charles University, Katerinska 32, 12000 Prague, Czech Republicpavol.klobusicky@helios.gesundheit.de (P.K.); 33rd Department of Inner Medicine, General University Hospital, U nemocnice 2, 12000 Prague, Czech Republic

**Keywords:** hypoglycemia, insulinoma, pancreas, Whipple’s triad

## Abstract

*Background and Objectives:* Insulinoma is a rare tumor of the Langerhans islets of the pancreas. It produces insulin and causes severe hypoglycemia with neuroglycopenic symptoms. The incidence is low, at about 1–2 per 1 million inhabitants per year. The diagnosis is based on the presence of Whipple’s triad and the result of a fasting test. Surgery is the treatment of choice. *Objectives:* A retrospective observational study of patients operated on for insulinoma in our hospital focused on the diagnosis, the type of surgery, and complications. *Materials and Methods:* We retrospectively reviewed patients operated on due to insulinoma. There were 116 surgeries between 2000 and 2022. There were 79 females and 37 males in this group. A fasting test and a CT examination were performed on all the patients. *Results:* The average duration of the fasting test was 18 h. Insulinoma was found in the body and tail of the pancreas in more than half of the patients. Enucleation was the most frequent type of surgery. Complications that were Clavien Dindo grade III or more occurred in 18% of the patients. The most frequent complications were abscesses and pancreatic fistula. Five patients had malignant insulinoma. *Conclusions:* Surgery is the treatment of choice in the case of insulinomas. The enucleation of the tumor is a sufficient treatment for benign insulinomas, which are not in contact with the main pancreatic duct. Due to the low incidence of the condition, the centralization of patients is recommended.

## 1. Introduction

Insulinoma is the most frequent functioning neoplasms of the gastrointestinal tract. It originates from the B cells of the islets of Langerhans. The insulinoma tumor was first described by Nicholls in 1902. Its incidence is low, at about 1–2 per 1 million inhabitants per year. Females are more likely to be affected than males (4:1). The tumor is usually solitary (90%), and multilocular microadenomatosis is rare. Most of these tumors are found in the pancreas (90%), and ectopic extrapancreatic localization is unusual. The majority of the tumors are benign, and fewer than 10% are diagnosed in malignant form. Their malignant potential is based either on aggressive spread or distant metastases. Two thirds of insulinomas are smaller than 2 cm and 30% are smaller than 1 cm. Malignant insulinomas are usually larger than 3 cm. Insulinoma can be a part of multiple endocrine neoplasia (MEN I), together with parathyroid and hypophyseal adenoma [1,2,3,4,5,6,7].

Patients with insulinomas suffer from recurrent episodes of hypoglycemia. The most frequent symptoms are neuroglycopenic (altered mental status, abnormal behavior, visual disturbances, etc.). Autonomic, adrenergic, and cholinergic symptoms are less frequent. The diagnosis of insulinomas can be challenging as patients are first examined in neurologic and psychiatric departments due to neuroglycopenic symptoms. A milestone in the diagnosis is the confirmation of Whipple’s triad; symptoms can be provoked by fasting or physical activity, low plasma glucose levels at the time of symptoms, and the immediate relief of symptoms after glucose administration [1,2,3,4].

A fasting test is considered the gold standard for diagnosis. Fasting provokes the clinical symptoms and leads to decreases in glucose levels. The test typically takes 72 h. Its sensitivity is 88.9% and its specificity is 100%. Biochemistry examinations show low plasmatic glucose levels (≤2.2 mmol/L), and high levels of insulin (≥6 mIU/L), proinsulin (≥5 pmol/L), and C peptide (≥0.2 nmol/L) [7,8].

The localization of the tumor can be difficult. The possibilities offered by imaging diagnostics have improved in recent years, especially with the improvement in CT and NMR resolution and the introduction of different types of PET CT diagnostics. These diagnostic tests are discussed later. Sometimes, preoperative localization fails, despite various techniques, and depends on the perioperative ultrasound and the manual examination of the pancreas by an experienced surgeon. This perioperative palpation of the pancreas has a sensitivity of 42–95% [5]. Occasionally, the tumor is found at a different site from where it was diagnosed preoperatively.

Surgery is the treatment of choice for insulinomas. Enucleation is indicated in smaller tumors without contact with the main pancreatic duct. The other possibility is a left-side pancreatectomy, with or without splenectomy. A central resection of the pancreas can be performed in some specific cases. A Whipple procedure or pylorus-preserving duodenopancreatectomy is indicated for larger tumors localized in the head of the pancreas, especially with suspected malignancy. The laparoscopic or robotic approach is suitable and increasingly used. The weakness of this minimally invasive method is that it prevents tactile perception, especially in patients in whom the localization of the tumor is uncertain.

Conservative treatment is possible with diazoxide (Proglycem) at a dose of 100–200 mg per day. Diazoxide reduces insulin secretion in tumor cells and influences gluconeogenesis [6]. However, it is effective in only approximately half of patients [7]. The other possibility is the use of octreotide to prevent further hypoglycemic attacks.

The prognosis of benign tumors is good, and there is no tendency toward recurrence. Malignant forms are rare, and their prognosis is generally much better than for other pancreatic tumors. The progression of malignant insulinomas is slow, and the metastatic form can be treated and significantly improved with long-acting somatostatin (octreotide).

Our goal is to retrospectively describe all the patients with insulinoma who were treated at our hospital over a 23-year period, with a specific focus on preoperative diagnosis and surgical treatment.

## 2. Materials and Methods

We retrospectively reviewed the records of patients undergoing insulinoma surgery over 23 years (2000–2022). Data were obtained from a prospectively maintained database.

Overall, we performed a total of 116 scheduled surgeries on patients suffering from insulinoma. The surgically treated postoperative complications are listed separately. There were 79 females, with a mean age of 52 years (20–83 years, median was 51 years), and 37 males, with a mean age of 53 years (29–81 years, median was 59 years). The duration of the symptoms varied from 1 month to 25 years (mean was 3 years). All patients were examined in the 3rd Department of the Inner Medicine General University Hospital (which is focused on metabolic disease), including insulinoma localization. A fasting test was performed on all patients during hospitalization. Additionally, the levels of calcium and phosphates in the serum were examined in all patients, as well as in the hypophyseal region, to exclude MEN I syndrome. Radiological examination depends on the diagnosis period, and CT scan was used in all cases (Figure 1 and Figure 2). This was due to the unavailability of MRI, especially in the first years of the study.

Patients were referred to surgery after confirmation of the diagnosis and localization of the tumor via a multidisciplinary team (surgeon, internist, radiodiagnostic, gastroenterologist).

## 3. Results

Organic hyperinsulinism was confirmed in all operated patients. The results of the fasting test are shown in Table 1. The mean duration of the fasting test was 18 h. After 24 h, 80% of the patients had completed the test. The glucose levels were lower than 2.5 mmol/L in all the patients at the end of the fasting test.

Insulinomas were found in the body and tail of the pancreas in 60% of the cases. (Table 2), and 17% had a localization in the pancreatic head. Diffuse multilocular adenomatosis was found in four cases. There were nine tumors found in one patient, but they produced different hormones. We were unable to find tumors in eight patients during surgery, but most of these patients underwent surgery in the first half of the study period. One of them underwent surgery twice—for enucleation of the suspicious tumor and, subsequently, for a left-side pancreatectomy, but both specimens were negative.

The approach to these patients was highly selective, depending on the severity of their symptoms, their age, comorbidities, etc. An explorative laparotomy with or without biopsy was performed in seven cases. We performed a left-side pancreatetomy in two cases. The reason for these “blind” resections was severe hypoglycemic attacks before surgery. But the pathological examinations of the specimens were negative in all these cases. One patient in this cohort refused further examinations and surgery. A second patient underwent surgery again due to symptoms, and a left-side pancreatectomy was performed but, unfortunately, the specimen was negative again. Two patients were advised to receive a second surgery after re-examination, leading to a new confirmation of the tumor and its localization. A left-side pancreatectomy was performed, and the insulinomas were found. Their size was 20 mm and 25 mm in diameter. The interval between both surgeries was 6 months. Unfortunately, one tumor was malignant, with lymph-node metastases. However, this patient is now not undergoing treatment, with no signs of recurrence, and she is alive 18 years after surgery. In this patient, MEN I syndrome was confirmed later. To summarize, in this group with a negative first surgery, in two cases, we found tumors during re-examination and successfully removed them. The rest of the patients were treated conservatively with diazoxide and they are without signs of the severe hypoglycemia.

The type of surgery performed depends on the tumor localization, its size, and its relationship to the main pancreatic duct and surrounding vessels and organs. Enucleation was performed on 50% of the patients. (Figure 3). In some cases, we added Roux-Y pancreaticojejunoanastomosis. The second most frequently used procedure was left-side pancreatectomy (37%), with or without splenectomy. Other types of surgery were indicated less frequently (Table 3). The laparoscopic approach was used on 33 patients, but conversion was necessary in 2/3 of these cases. The main indication for conversion was uncertainty about the localization of the tumor. Five patients underwent surgery twice, and one female was underwent surgery four times (three of these patients are mentioned above.

Tumor enucleation was achieved for one patient during their first surgery. This patient maintained a normal glucose level after surgery, but 4 years later, the symptoms of hypoglycemia returned, and a new tumor was found. Left-side pancreatectomy was performed, and the patient is now without any signs of hyperinsulinism. The last patient in the cohorts received a left-side pancreatectomy during their first surgery. The pathologist confirmed diffuse nesidioblastosis, and, due to the persistence of hypoglycemia, the patient was advised to receive a pancreatectomy two years later. The result of the pathological examination was the same—diffuse nesidioblastosis. The patient who received four operations started with a left-side pancreatectomy without splenectomy. Insulinoma was confirmed. Three years later, the symptoms recurred, and a tumor in the spleen hilum was diagnosed. A splenectomy was performed, and insulinoma was confirmed by a pathologist. Two years later, the same situation arose: symptoms of hyperinsulinism were observed and a tumor was found near the previous resection line. This tumor was removed and insulinoma was confirmed again; however, it was a malignant form, because there were lymph-node metastases present. The recurrence of the hyperinsulinism appeared six years later, and the tumor was found in the head of the pancreas. A subtotal resection was performed and, again, a tumor and diffuse nesidioses were found. The patient is alive 16 years after the first surgery, without any signs of insulin hyperproduction, or of the recurrence or metastases of malignant insulinoma

Postoperative complications occurred in 38% of the patients. Serious complications, i.e., of Clavien Dindo grade III and higher, occurred in 18% of the patients (Table 4). Most of these were subphrenic abscesses and pancreatic fistula (2/3 of all the complications). Reoperations were performed in eight cases. One patient died postoperatively, and this patient suffered from advanced malignant insulinoma and died due to bronchopneumonia and cardiac failure.

If we compare the first (before 2011) and second (2011 and later) decades in our series, there was a clear trend towards more complex procedures. Resections prevailed over enucleations. In both groups, we started with laparoscopy in a third of the operations. The percentage of complications did not change (Table 5).

Malignant insulinoma was diagnosed in five cases (two females and three males). The diagnosis was confirmed pathologically. The confirmation of the malignancy was based on the aggressive infiltrative growth of the tumor, or the presence of metastases in the lymph nodes or in the liver. Malignant insulinomas were localized in the body of the pancreas in two cases, as mentioned earlier. Malignant insulinomas were found in the head of the pancreas in two cases, of which one was found in the uncinate process. One patient died postoperatively, while two survived and lived 5 and 6 years respectively, and two are still alive 10 and 16 years after surgery. The details of all the cases are summarized in Table 6.

We diagnosed MEN I syndrome in three patients. Two of them were family members (mother and daughter). The daughter was the patient with nine different pancreatic endocrine tumors. Her history started with a hypophyseal tumor producing prolactin 10 years before her surgery for insulinoma, which was treated by irradiation. The parathyroid adenoma was found four years later, after the pancreatic surgery, and was treated by surgery. There is another member of this family with MEN I syndrome, but she is not included in this group of patients because she underwent surgery outside of the interval of 2000–2022.

The third patient with MEN I syndrome had insulinoma first, and other tumors developed nine years (parathyroid tumor) and eleven years (hypophyseal tumor) later, respectively. This patient is one of the living patients with malignant insulinoma. Genetic screening found only a MUTYH (MutY DNA glycosylase) mutation. One of our patients is now under examination due to the suspicion of a parathyroid tumor.

## 4. Discussion

Proper diagnostic and therapeutic management of insulinomas are based on a multidisciplinary team consisting of an endocrinologist, a pancreatic surgeon, radiologists, and endoscopists [4].

There are some difficulties in the diagnosis and treatment of insulinomas, despite improved diagnostic options.

Firstly, due to neuroglycemic symptoms and the uncommon occurrence of the disease, physicians often do not consider this diagnosis [7]. On the other hand, the confirmation of the diagnosis is based on the presence of the Whipple’s triad. The median time from the onset of clinical signs to diagnosis is about two years [3,7]. The time between the onset of the symptoms and accurate diagnosis in our group was longer, about 3 years. The longest interval was 25 years. In these cases, with long intervals to accurate diagnosis, is there a high risk of irreversible brain injury due to repeated hypoglycemia. On the other hand, the shortest interval to diagnosis in our patients was one month. A serious differential diagnostic problem could be hypoglycemia factitia, due to the similarity between the laboratory-test results for this condition and those in cases of insulinoma. This form of hypoglycemia is self-induced, through the administration of insulin or oral hypoglycemic drugs [7].

It is necessary to exclude MEN I syndrome after diagnosis. As with the other procedures used in this long study, the approach to ruling out this syndrome evolved according to current guidelines and to the availability of individual diagnostic methods. However, biochemical tests and a CT scan of the pituitary gland were used for all the patients. Of course, we consider MRI to be a better method, but its availability was initially lower. However, the tumors can occur one by one, as well as simultaneously. In our two patients with MEN I syndrome, insulinoma occurred first and second, respectively, in the history of their disease.

The treatment of choice is surgery. Therefore, it is necessary to establish the precise localization of the tumor. The sensitivity and specificity of radiodiagnostic methods are rising. A conventional CT scan has a sensitivity of about 33%, while that of a single-slice helical CT is 58%, and that of multidetector CT is 75–100% [8]. Insulinomas are hypervascular, so they have greater enhancement during the arterial phase of the examination. The can be said of MRI. A conventional MRI can detect insulinoma in about 31% of cases, and multiphasic MRI can detect insulinoma in 85% of patients [4]. The use of 18 F-fluorodeoxyglucose (18 F-FDG) PET is not helpful in diagnosis due to the low proliferative activities of these tumors. Furthermore, 68 Ga-DOTApeptide PET is also not helpful, because it shows a low expression of SSTR subtype 2 [1]. The 68 Ga-DOTATATE scan is more sensitive for localization [1], but according to other studies, its sensitivity is lower than 20% [9]. Recent data show that 68Ga-DOTA-Exendin-4 PET/CT has better results, with a sensitivity of nearly 85% and a specificity of 100%. Exendin is a synthetic glucagon-like peptide-1, which is expressed in cases of insulinoma [10]. Unfortunately, we do not have personal experience with this method. Contrast ultrasound and endoscopic ultrasound are highly sensitive and specific (90–100%) [1]. All these methods are complementary, not competitive [4]. Despite the progress in diagnostic methods, in some cases, surgery can be indicated without precise knowledge as to the position of the tumor. An experienced surgeon can find the tumor in 42–95% of cases [1,5,11]. Moreover, there are some case reports about the intraoperative near-infrared imaging of insulinomas in the literature [12,13].

The only curative treatment for insulinomas is surgery. Surgical cure rates range from 77% to 100% [14]. The type of surgical treatment depends on the localization and size of the tumor. Usually, enucleation and left-side pancreatic resection are the treatments of choice. Nearly 90% of patients can be treated with this type of surgery [2,4,15,16]. The same was seen in our group of patients. The recurrence rate is about 7%, but it is higher in MEN 1 patients [8]. We found recurrences in three of our patients, which is lower than the rates in the literature. The intervals between surgeries were 9 and 12 years. The third case is described above (malignant insulinoma, repeat surgeries).

Contemporary clinical guidelines for neuroendocrine tumors are published in https://www.enets.org/guidelines.html (accessed on 18 July 2023) [17].

The laparoscopic approach has been increasingly applied. The best indication for this is a superficial tumor not in contact with the pancreatic or bile ducts or vessels. The conversion rate is about 17% [16]. According to our experience, the laparoscopic approach is suitable for precisely localized, smaller tumors. The most common reason for conversion in our group was difficulty in tumor identification and the lack of tactile perception. Overall, the conversion rate was much higher in our series.

The complication rate was quite high, at about 45–52%, as in other types of pancreatic surgery [2,15,16]. Surprisingly, there was no difference between the enucleation group and the pancreatic-resection group regarding the complication rate [15,18]. The complication rate in our series was lower. We attribute this lower percentage of complications to the number of patients admitted to our hospital and our many years of experience with this disease.

Malignant insulinomas are rare, and, as a result, only case reports and studies on small patient groups are present in the literature. The incidence of malignant insulinomas is described as 5–14% of all insulinomas [2,14,19,20]. The five-year survival is 24–55% [19,20]. The incidence of malignant insulinoma in our group was 4.3%, but four of our five patients survived for at least five years following diagnosis. The male/female ratio in cases of malignant insulinomas differed compared to the benign form, but this could have been caused by the small numbers studied.

The main weakness of our study was the long interval during which the patients were enrolled. Over 23 years, the guidelines for the diagnosis and treatment of neuroendocrine tumors have changed several times, the methods of diagnosis, especially imaging methods, have expanded significantly, and new biochemical tests have appeared. It is difficult to compare the availability of individual diagnostic methods at the beginning of the millennium and today. The pathological diagnostic options are also significantly different in 2023 from those in 2000. For these reasons, it is difficult to compare findings and procedures in the first patients and the current patients. On the other hand, this long interval, with different patient-management options, can also be considered a strong point of our study. It shows that experienced teams with a strong focus on the diagnosis and treatment of rare diseases can achieve success even with insufficient methods for diagnosis (from the current point of view).

In the future, we can expect more precise approaches to the pre- and peri-operative localization of these tumor and, probably, the rapid spread of the minimally invasive approaches, especially robotic methods, supported by near-infrared navigation or another technique for better visualization of pathological tissues.

## 5. Conclusions

Overall, a multidisciplinary approach to patients with organic hyperinsulinism provides the best results. Surgical treatment remains the gold standard for the management of insulinomas. The primary goal of surgery is to remove the tumor while preserving the surrounding pancreatic tissue and maintaining normal endocrine function. The success of the surgery depends on the accurate preoperative localization of the tumor. The centralization of diagnostic and treatment capacities is recommended due to the sporadic incidence of insulinomas, and it generally ensures better treatment results.

## Figures and Tables

**Figure 1 medicina-59-01423-f001:**
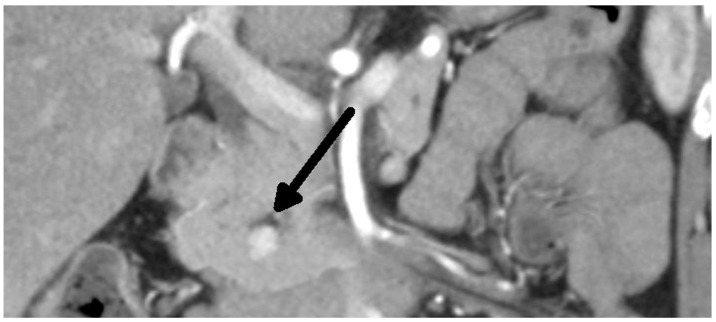
Insulinoma—CT arterial phase (tumor marked with an arrow).

**Figure 2 medicina-59-01423-f002:**
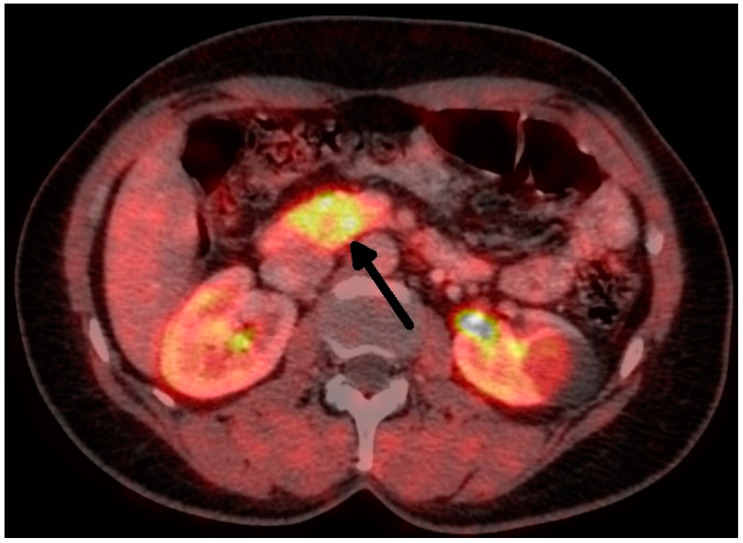
Insulinoma—FDOPA PETCT (tumor marked with an arrow).

**Figure 3 medicina-59-01423-f003:**
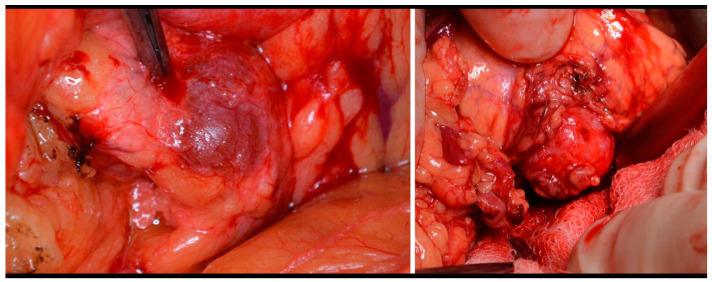
Insulinoma—perioperative view and enucleation.

**Table 1 medicina-59-01423-t001:** Results of the fasting test.

	Start (Mean ± SD)	End (Mean ± SD)
Glucose (mmol/L)	3.4 ± 1.4	1.7 ± 0.4
Insulin (mU/L)	54 ± 36	56 ± 46
Insulin/glucose	18.9 ± 15.3	32.2 ± 30.3
C peptide (nmol/L)	1.11 ± 0.61	1.15 ± 0.74

**Table 2 medicina-59-01423-t002:** Tumor localization.

Localization	No. of Patients (%)
Head	20 (17%)
Uncinate process	8 (7%)
Neck	4 (3%)
Body	38 (33%)
Tail	32 (28%)
Diffuse	4 (3%)
Multiple tumors (9)	1 (1%)
Tumor not found	9 (8%)

**Table 3 medicina-59-01423-t003:** Type of surgery.

Surgical Procedure	Benign Insulinoma (*n* = 111)	Malignant Insulinoma (*n* = 5)
Exploration	6 (5%)	1
Enucleation	49 (45%)	
Enucleation + Roux Y pancreaticojejunoanast	9 (8%)	1
Lef-side pancreatectomy	16 (14%)	2
Left-side pancreatectomy + splenectomy	27 (24%)	
Central resection	1 (1%)	
Head resection	1 (Beger procedure) (1%)	1 (Whipple procedure)
Subtotal pancreatectomy	1 (1%)	
Total pancreatectomy	1(1%)	

**Table 4 medicina-59-01423-t004:** Complications, Clavien Dindo grade III+ (*n* = 21).

Subphrenic abscess	8 (38%)
Pancreatic fistula	7 (33%)
Acute pancreatitis	3 (14%)
Burst abdomen	1 (5%)
Bleeding arteria lienalis	1 (5%)
Death	1 (5%)

**Table 5 medicina-59-01423-t005:** Comparison of the first and second decades.

	2000–2010 (*n* = 57)	2011–2022 (*n* = 59)
Male:female	16:41	21:38
Mean age	51.5 (20–81)	53.2 (21–83)
Localization		
Right side (head, neck, uncinatus)	16 (29%)	14 (24%)
Left side (body, tail)	32 (57%)	38 (66%)
Diffuse, multiple	3 (5%)	2 (3%)
None found	5 (9%)	4 (7%)
Type of Surgery		
Enucleation	38 (68%)	18 (31%)
Resection	14 (25%)	36 (62%)
Other	4 (7%)	4 (7%)
Laparoscopic approach	19 (34%)	19 (33%)
Complication	13 (23%)	14 (24%)

**Table 6 medicina-59-01423-t006:** Malignant insulinoma.

Male/ Female	Age	Localization	Size (mm)	Surgical Procedure	Signs of Malignancy	Pathology	Follow Up
F	61	Head	10	Excision + Roux Y	Infiltrative growth	G1 1/10 HPF	Postoperative death
F	43	Body	25	Left-side pancreatectomy + splenectomy	Lymph-node metastases	G1 1/25 HPF	Alive 11 years after surgery
M	40	Body	25	Left-side pancreatectomy + splenectomy	Lymph-node metastases	G1	Allive 18 years after surgery
M	64	Head	30	Exoloration	Liver metastases	G1	+ 6 years after surgery (multiorgan falure due to severe acute pancreatitis)
M	77	Proc. uncinatus	35	Whipple	Liver and lymph-node metastases	G2	+ 5 years after surgery (cardiac failure, but liver metastases at the time of the death)

HPF—High-power fields.

## Data Availability

Data available on request due to restrictions e.g., privacy or ethical. The data presented in this study are available on request from the corresponding author. The data are not publicly available due to that informed consent of patients does not allow publication of datasheet.

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
