# Peer review of "Diagnosis and Surgical Management of Insulinomas—A 23-Year Single-Center Experience"

_medicina, 2023, doi:10.3390/medicina59081423_

Round 1
Reviewer 1 Report
The topic of the study is highly relevant and carries substantial importance within the field. The research question being addressed has the potential to contribute valuable insights and advance scientific knowledge. The manuscript's potential impact on clinical practice or the broader research community is noteworthy.
However, one of the primary concerns with this manuscript lies in its level of presentation. The overall quality of English in the manuscript is poor, which significantly affects the readability and comprehension of the study. It is crucial for the authors to invest time in improving the language, grammar, and sentence structure to ensure clarity and coherence throughout the manuscript. Consider seeking assistance from a professional editor proficient in scientific writing to rectify these issues.
Furthermore, the manuscript would benefit from a more robust statistical analysis.
Another area for improvement is the need for more comprehensive data. While the study holds promise, the available data seems insufficient to draw firm conclusions. It is important to consider expanding clinical information, visualizing technics that were used and give data on follow up if possible.
Finally, I would encourage the authors to discuss the limitations of their study. Acknowledging and addressing potential limitations, such as sampling biases, confounding variables, or any other factors that may affect the validity of the results, will provide a more comprehensive and balanced interpretation.
See my suggestions
Line 46 – check current guidelines for diagnosis of organic hyperinsulinsm (ratio is not used anymore)
Line 54 – not clear what authors are trying to say
Line 74 – what does mean average age? Is it mean or median? Please indicate demographics according to basic stats (Mean±SD or Me [25-75%] + range)
Line 76 – what is 3d department? Endocrinology dept? Not clear
Line 63 - would be good to add data on other possible treatment (long acting somatostatin analogues and everolimus for malignant cases)
Line 79 – please give more details on visualizing methods used (what was done? in how many cases)
Line 80 – was the genetic testing for MEN1 gene performed? If yes, in how many cases?
Line 100 – how can you be sure that it is insulinoma if the tumor was not found?
Line 124 – no data on histopathology and Grade level. According to what malignant insulinoma was diagnosed? Were there distant metastases? How old were patients with malignant tumors? Did they have multiple lesions and what was the size of insulinomas?
Line 143 – why CT, but not MRI of the pituitary was preferred?
Line 153 – please add data on Exendin PET
Line 198 – please correct
The overall quality of English in the manuscript is poor, which significantly affects the readability and comprehension of the study. It is crucial for the authors to invest time in improving the language, grammar, and sentence structure to ensure clarity and coherence throughout the manuscript. Consider seeking assistance from a professional editor proficient in scientific writing to rectify these issues.
Author Response
Thanks for the review with the reminders and I've dealt with it as best as I could.
Language proofreading was done, clinical information about patients and details about disease groups were added. The discussion was also expanded. Individual comments on the text have been modified. The 3rd Department of the Inner Medicine is one of the clinics in our hospital where all patients were examined. One of the authors works here and treats patients. This is the official name of the department. Data on Exendin PET have been added, although we have no experience with this method. The use of MRI for examination was limited by the poor availability of this method (this has, of course, already changed)
I tried to solve all other comments in the text, which is practically completely changed.
Reviewer 2 Report
- A more detailed descriptive analysis of the clinical features of the patients at the time of diagnosis is needed.
- Patients who required more than one surgery should be described.
- Quantitative variables should be expressed as mean +/- standard deviation or median (interquartile range) where appropriate.
- Were other causes of hyperinsulinism ruled out in patients in whom the tumor was not found at surgery. Were other diagnostic tests performed, e.g. calcium selective intra-arterial stimulation test, intra-operative pancreatic ultrasound, 68Ga-Exendin-4 PET/CT, etc.
- Diagnostic and therapeutic procedures in cases of malignant insulinoma are not described.
- How were patients with a suspected occult insulinoma managed?
- The discussion is merely descriptive of the findings obtained from the retrospective study.
- The conclusions of the study are very general and unspecific.
- English and the general wording of the manuscript should be revised.
- English and the general wording of the manuscript should be revised.
Author Response
Thanks for the review with the reminders and I've dealt with it as best as I could.
I added more clinical information especially in the groups where no tumor was found and also in the malignant insulinoma group. A new table was also added here. The text has been completely revised and expanded. A language correction was also made
I tried to solve all other comments in the text, which is practically completely changed.
Round 2
Reviewer 1 Report
The manus improved, but still requires corrections
1. Please check once again the quality of translation – a lot (!!!!) of typos and grammar mistakes and awkward phrases (some are highlighted through the text)
2. Statistics in the results – please indicate what is average (Mean or median???) lines 86-88 and 102
3. Table 1 – results are given as Means and +- SD I guess, but according to the values reconsider the use of medians and Q25-75%
4. Line 117 – patient with 9 different NETs - was he screened for MEN1 syndrome?
5. Please give results in how many patients you found PTH or pituitary adenomas?
6. Lines 122-135 – very confusing description!!! Please simplify and restructure.
7. 9 cases in whom insulinoma have not been found – how can you claim that it was insulinoma???
8. Table 3 – please indicate in columns ‘n’
9. Line 141 – 37% reflects the total amount of left side pancreatectomy or only those without splenectomy? It is not clear now
10. Table 4 – again add `n` to the title
11. Table 5 – what is HPF? Please give legend below the table. What was G level in patient 4? Did patients 4 and 5 died 6 and 5 years postop or they were lost for follow up? Please clarify and in case of death - provide more data
12. Lines 215-220 – please indicate whether these results are from your cohort or taken in the literature?
13. Line 221 – have you seen Munchausen syndrome among your patients with hyperinsulinemic hypoglycemia? Please compare your own results with literature data.
14. Line 224 – could you please clarify whether genetic testing for MEN1 syndrome was performed in any of you patients?
15. Line 231-250 – please compare your own results with literature data
16. Line 279 – you mention Octreotide in conclusions only, although there was no information about it through though the manuscript….
17. Line 309 – correct!!!!!!!
Requires correction
Author Response
Dear reviewer
Thank you again for your time what you spend with the reading and your proposal how to improve my manuscript.
I tried my best to deal with your comments.
I checked and correct the typos. The English was corrected by the MDPI professional service. I improved statistical terms and tables according to your recommendation.
I added the information about MEN I syndrome.
About Munchausen syndrome, we have seen one patient where is probably this syndrome diagnosed, but it was in 90tie, so she is outside of this group of patients. I have mentioned this only in differential diagnostics.
I think that it is impossible to compare the results of imaging examination and the literature because during the period of 23 years our CT, MR and PET CT devices was repeatedly changed and improved. Moreover, when the patients came with clear finding o CT or MR what was performed outside of our hospital, we neither did repeat it.
Thank you very much and I am looking forward to your reply.
Reviewer 2 Report
.
.
Author Response
Dear reviewer
Thank you again for your time what you spend with the reading and your proposal how to improve my manuscript.
I tried my best to deal with your review.
I checked and corrected the typos. The English was corrected by the MDPI professional service. I improved statistical terms and tables according to your recommendation.
There were added some more information’s about MEN I patients and malignant insulinomas.
Thank you very much and I am looking forward to your reply.